# A bioelectronic device for electric field treatment of wounds reduces inflammation in an *in vivo* mouse model

Cristian O. Hernandez[1]☯, Hao-Chieh Hsieh[1]☯, Kan Zhu[2,3], Houpu Li[1], Hsin-ya Yang[2], Cynthia Recendez[2,3], Narges Asefifeyzabadi[1], Tiffany Nguyen[1], Maryam Tebyani[1], Prabhat Baniya[1], Andrea Medina Lopez[2], Moyasar A. Alhamo[2], Anthony Gallegos[2], Cathleen Hsieh[1], Alexie Barbee[1], Jonathan Orozco[1], Athena M. Soulika[2,4], Yao-Hui Sun[2,3], Elham Aslankoohi[1], Mircea Teodorescu[1], Marcella Gomez[5], Narges Norouzi[6], Roslyn Rivkah Isseroff[2,7]*, Min Zhao[2,3]*, Marco Rolandi[1]*

1 Department of Electrical and Computer Engineering, University of California, Santa Cruz, CA, United States of America, 2 Department of Dermatology, University of California, Davis, Sacramento, CA, United States of America, 3 Department of Ophthalmology & Vision Science, University of California, Davis, Sacramento, CA, United States of America, 4 Pediatric Regenerative Medicine, Shriners Hospitals for Children, Sacramento, CA, United States of America, 5 Department of Applied Mathematics, University of California, Santa Cruz, CA, United States of America, 6 Department of Electrical Engineering and Computer Sciences, University of California, Berkeley, CA, United States of America, 7 Dermatology Section, VA Northern California Health Care System, Mather, CA, United States of America

☯ These authors contributed equally to this work.
* mrolandi@ucsc.edu (MR); minzhao@ucdavis.edu (MZ); rrisseroff@ucdavis.edu (RRI)

**Data Availability Statement:** All relevant data are within the manuscript and its Supporting Information files.

## Abstract

Electrical signaling plays a crucial role in the cellular response to tissue injury in wound healing and an external electric field (EF) may expedite the healing process. Here, we have developed a standalone, wearable, and programmable electronic device to administer a well-controlled exogenous EF, aiming to accelerate wound healing in an in vivo mouse model to provide pre-clinical evidence. We monitored the healing process by assessing the re-epithelization rate and the ratio of M1/M2 macrophage phenotypes through histology staining. Following three days of treatment, the M1/M2 macrophage ratio decreased by 30.6% and the re-epithelization in the EF-treated wounds trended towards a non-statically significant 24.2% increase compared to the control. These findings provide point towards the effectiveness of the device in shortening the inflammatory phase by promoting reparative macrophages over inflammatory macrophages, and in speeding up re-epithelialization. Our wearable device supports the rationale for the application of programmed EFs for wound management *in vivo* and provides an exciting basis for further development of our technology based on the modulation of macrophages and inflammation to better wound healing.

## Introduction

Wound healing involves several important signaling processes [1–4], with a critical one being the electric field (EF) [5–10]. An injury in the skin generates endogenous EFs driven by ion

**Funding:** This project is supported by the Defense Advanced Research Projects Agency (DARPA) through Cooperative Agreement Number D20AC00003 awarded by the U.S. Department of the Interior (DOI), Interior Business Center. There was no additional external funding received for this study.

**Competing interests:** The authors have declared that no competing interests exist.

movement across epithelial layers, guiding directional cell migration either in the cathodal or anodal direction, depending on cell type [5, 6, 11–14]. Researchers have used external EF stimulation to accelerate wound closure [15–18] by enhancing cell migration [19] and modulating cellular phenotypes [20]. During healing, different macrophage phenotypes contribute to immune and inflammatory responses critical for repair, and modulating these responses with EF is important for understanding this modality affects wound healing. The broad simplified phenotypic category of the M1 macrophages promotes inflammation, while M2 macrophages promote tissue regeneration [21–25]. Here, we use M1 and M2, which are much simplified descriptions, but suffice to illustrate our results. To deliver EF in vitro and in vivo, systems typically require several sub-components, including a power supply, voltage regulation, and a programmable controller [16, 26, 27]. In addition, the EF wound treatment system requires an interface with the tissue, such as a salt bridge or hydrogel system to convert the electron current in the instrument into ionic current in the wound [26, 28]. Here, we designed and fabricated a wearable, compact and battery-powered EF delivery device and tested its efficacy in a mouse wound model (Fig 1A). We integrated the whole EF delivery setup into a single device without the need for an external connection, which includes a battery-powered micro-controller unit as the programmable voltage regulator and a reservoir with a hydrogel as the interface

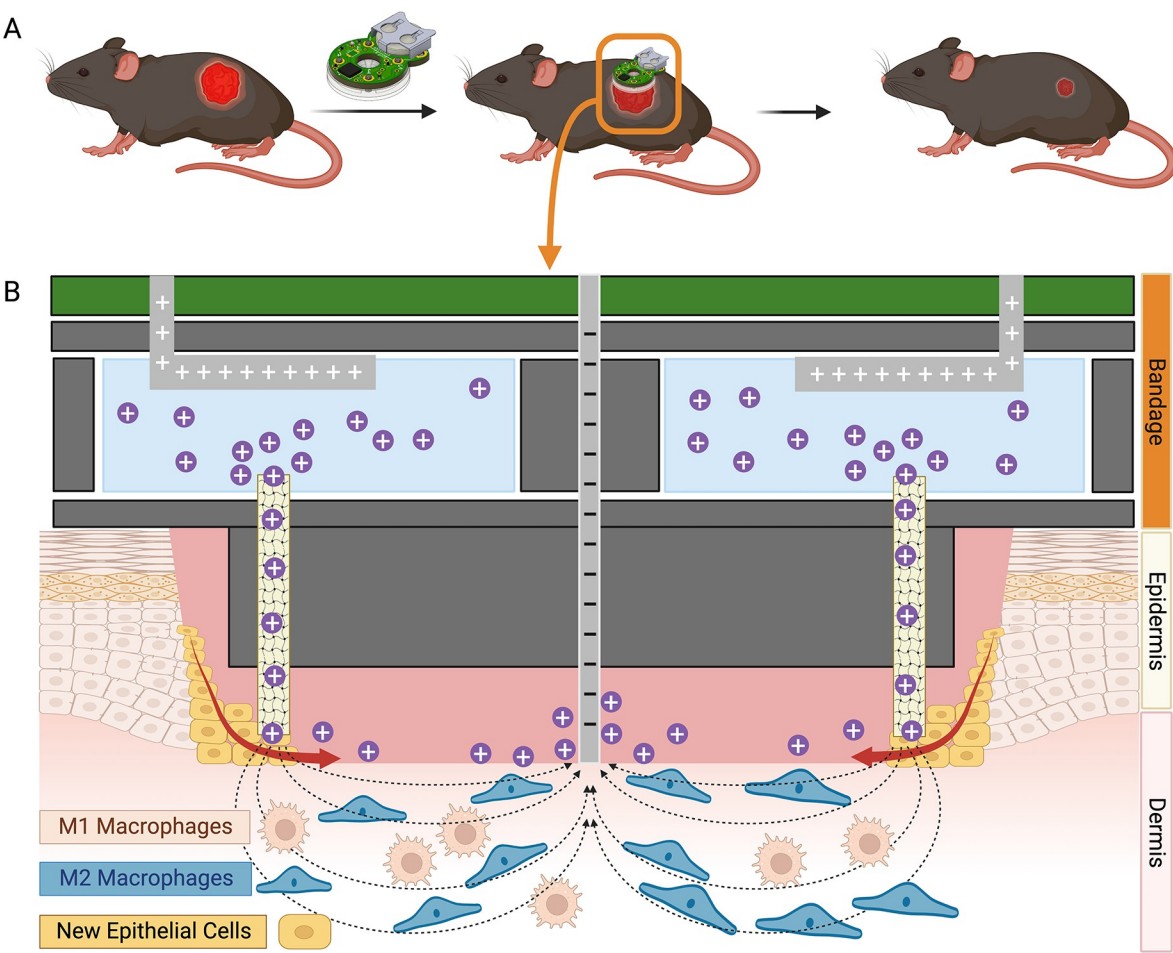

**Fig 1. Macrophage phenotype, wound schematic, and endogenous electric field at mouse wound. (A).** A battery-powered bioelectronic device for EF delivery was applied to the mouse wound. **(B)** the EF delivery device generates EFs in the wound through the hydrogels and electrode, pointing from wound edge to center. Macrophages and new epithelial cells would be modulated in the healing process.

to the wound (Fig 1B). The device allows us to deliver EF to the wound in a programmable manner with spatio-temporal control with minimal human interactions with the animals and monitor its effects on re-epithelization and macrophage polarization.

## Results and discussion

The wearable bioelectronic device expedites murine wound healing via programmable electric field stimulation and is comprised of two modules: (1) a programmable battery-powered controller unit and (2) an electric field delivery stimulation unit (Fig 2). Module 1 consists of a battery, microcontroller unit, and printed circuit board (Fig 2A). Module 2 consists of metallic contact pins, polydimethylsiloxane body, silver chloride (AgCl) wires, Steinberg solution, and a hydrogel to interface with the tissue (Fig 2A). The microcontroller can be programmed to output voltage to 4 channels (Fig 2B). To control electrical stimulation, the low-power microcontroller unit is programed to apply 3V for 250-minutes daily (Fig 2C). The microcontroller enters deep-sleep mode to prolong battery run-time once the 250-minute treatment ends (Fig 2C). The EF delivery schematic outlines how module 2 delivers an EF of 125 ± 75 mV/mm to the wound during actuation (Fig 2D). We designed our bioelectronic device to emit an electric field strength of 125 ± 75 mV/m because prior studies indicate that the endogenous electric fields at skin and cornea wounds of rodents typically range from 100 to 200 mV/mm [29, 30]. We aim to enhance the electric field at a skin wound to a range that has been proven to be biologically compatible and conducive to promoting directional cell migration and wound healing without causing harm to cells and tissues [31]. We iterated through various version of microcontroller design with increasing degree of complexity (S1 Fig in S1 File) and for this work we used version 2. We performed quality control using a digital multimeter (S2 Fig in S1 File) on our bioelectronic device after full assembly fabrication (S1 Table in S1 File), and during in vivo experiment (S2 Table in S1 File). We select the devices with all four working channels that can deliver the desired EF strength that ranges from 200 mV/mm to 125 mV/mm (see calculations in supporting information). The device EF strength was affected by resistance generated from the Ag/AgCl electrodes, hydrogel capillaries, and the wound resulting in a voltage drop throughout the circuit system (S3 Fig in S1 File). The hydrogel-filled capillaries function similarly to the salt bridge utilized in prior *in vitro* experiment serving as a conducive interface [18], establishing direct contact between the device and the wound bed for efficient bioelectronic transmission of the EF. The Ag reference electrode serves as a reference point for the electric potential. The potential difference creates an electric gradient that extends from the active hydrogel-filled capillaries electrodes towards the reference electrode, generating the desired lateral EF that establishes a 125 ± 75 mV/mm EF in the wound pointing from edge to center. The four capillaries are located closest to the wound edge, with the reference electrode making contact at the center of the wound (Fig 2E), generating a directional EF with the anode at the wound edge and cathode at the center. The electrode layout generates an EF that delivered stimulation to promote wound healing (Fig 2E).

Before performing in vivo work, we performed *in vivo* biocompatibility testing to ensure that all of the components of the device would not affect the mouse or the wound healing process (S3 Table in S1 File). All materials that come in contact with the wound resulted in no adverse effect during biocompatibility testing. For example, the volume of the hydrogel including all four capillaries equates to approximately 0.157 μL. The volume used in the device is negligible thus, rendering hydrogel unlikely to induce toxicity or an enrichment effect that can aid in healing. We performed experiments in a mouse wound model and delivered an EF treatment for 250-minutes daily for 3 days (Fig 3). We decided to focus our analysis on re-epithelialization and macrophage phenotype. Histological assessment of wound epithelialization is a

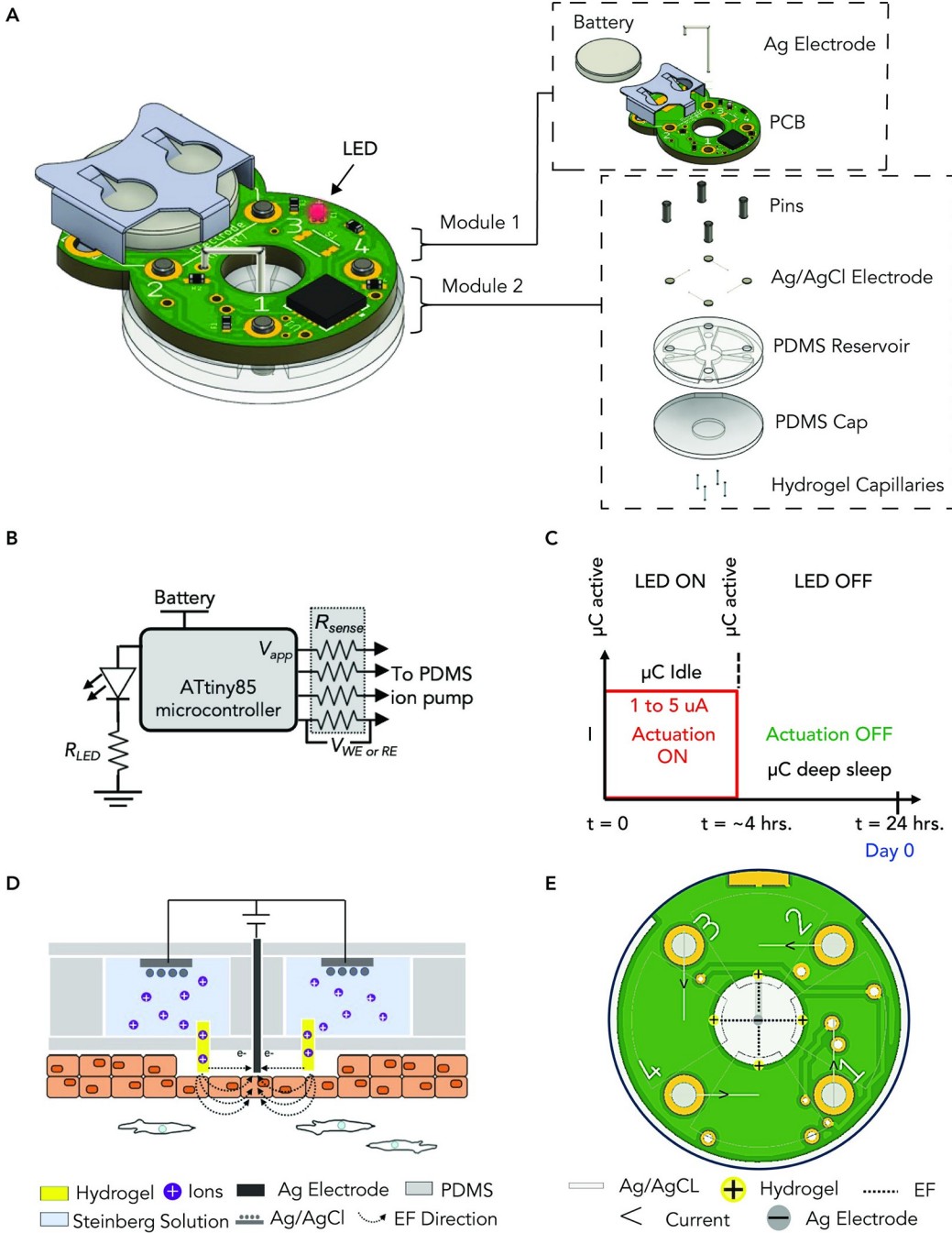

**Fig 2. *In vivo* bioelectronic device that delivers tuned electric field for wound treatment. (A)** In this setup, the external power source generates a potential difference between the reservoir embedded Ag/AgCl electrodes. This potential difference drives the flow of electric current through the electrolyte, Steinberg solution, filling the reservoirs. The electrolyte, being a conductive solution, facilitates the movement of ions between the hydrogel-filled capillaries. The hydrogel-filled capillaries function similarly to the salt bridge utilized in the *in vitro* experiment. The hydrogel-filled capillaries serve as a conducive interface, establishing direct contact between the device and the wound bed for efficient bioelectronic transmission of the electric field. The Ag reference electrode serves as a reference point for the electric potential. The potential difference creates an electric gradient that extends from the active hydrogel-filled capillaries electrodes towards the reference electrode, generating the desired lateral electric field that establishes a 125 ± 75 mV/mm EF in the wound pointing from edge to and converging at the wound center. **(Right Panel of A)** Exploded view detailing the components that make up the modules of the bioelectronic device. **(B)** Schematic of the circuit. **(C)** Actuation of electric field treatment. LED on indicates treatment is actuating. After 250-minutes the treatment stops and goes on sleep mode. **(D)** Schematic diagram of the electric circuitry on a wound. **(E)** Bottom view of device and how it generates the electric field.

more definitive measurement of healing compared to evaluating the wound surface because the new epithelial tongue is just a few cells deep and cannot be visualized macroscopically [32, 33]. Surface images of wound edges are often obscured by eschar or the inability to focus sharply on different planes simultaneously on a living animal's curved dorsum [32, 33]. After the described treatment, the wound tissue was excised and examined for wound re-epithelialization and for macrophage phenotype (Fig 3A).

Although macrophages comprise a large spectrum of phenotypes, for simplicity here we use the dominant simplified nomenclature in the literature of M1 (pro-inflammatory subtype) and M2 (pro-reparative subtype) [6, 23–25, 34, 35]. F4/80 was used as a pan-marker to label all macrophages present in the wound and iNOS expressing M1-like macrophages as well as CD206 expression M2-like macrophages were detected and counted as a fraction of the total F4/80+ macrophages present in the wound. The ratio of M1/ M2 macrophages was calculated for electric field-stimulated and control wounds in sections of the wound center at day 3 post -wounding. Representative images demonstrate that macrophages of both phenotypes were observed in the wound center of control wounds with the iNOS+F4/80+ M1-like cells being dominant (Fig 3A). Our histology results provide further representative images of re-epithelialization with green markings that indicate the new re-generated layer (Fig 3B). We have provided a closer look on the new re-generated layers in our supplementary information (S4 Fig in S1 File). In the wounds treated with our EF device, the percentage of M1 macrophages was slightly reduced while that of the M2-like cells showed a tendency of increase when compared to the non-EF control (S5 Fig in S1 File). However, the M1/M2 ratio was significantly reduced by 30.6% after EF treatment (Fig 3C).

The results suggest that EF stimulation to mouse wounds may be promoting the healing process by altering macrophage phenotypes, through the reduction of inflammatory M1 phenotype macrophages, thereby decreasing inflammatory stimuli and promoting earlier wound healing, such as a rapid granulation and the accompanying tissue generation, by increasing the relative presence of reparative M2 macrophages. Our previous *in vitro* research indicates differential electrotactic responses in macrophages pre and post-Salmonella infection, suggesting phenotype influence [28]. This highlights the potential to modulate healing processes through applied EFs [28]. A different *in vitro* study demonstrated that EF generated by the group's bioelectronic device could facilitate the polarization of cells of the THP-1 human leukemia monocytic line, used to study monocyte/macrophage function, toward the M2 phenotype, specifically M0 to M2 [36]. However importantly, the macrophage phenotype and relative percentage in wounds *in vivo* during external EF stimulation is underexplored. Our results examine this very important outcome of EF stimulation and support new emerging evidence that indicates EFs can induce macrophage polarization of M0 to M2 [28, 36–38]. Moreover, the histomorphometric analysis of the treated and control wound beds indicated a trend towards an increase in wound edge re-epithelization (Fig 3D). There was a 24.2% (p-value of 0.09) increase in re-epithelization in the electric field treated wounds relative to the controls. The predominance of M2 pro-reparative macrophages in the treated wounds (S6 Fig in S1 File) may be contributing to this epithelial proliferative phenomenon, as has been demonstrated in other tissues [39].

## Conclusions

Our study introduces a standalone, wearable, and programmable device for $E_f$ stimulation of wounds *in vivo*. In a murine model, the device is well tolerated by the animal and succeeds in reducing inflammation. This reduction in inflammation was measured with the reduction of M1/M2 ratio by 30.6%. Additionally, re-epithelization trended towards 24.2% suggesting

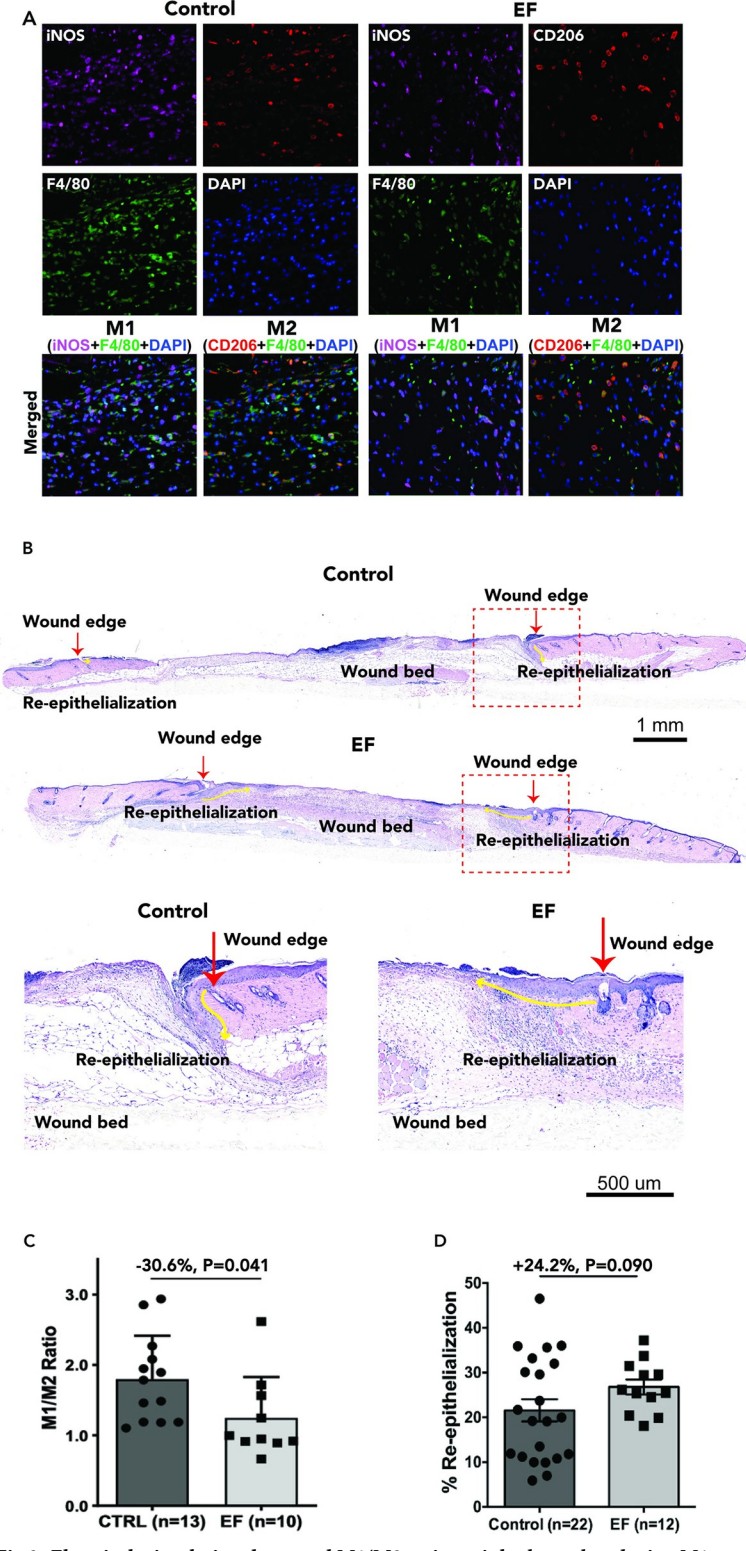

**Fig 3. Electrical stimulation decreased M1/M2 ratio mainly through reducing M1 marker-positive macrophages during wound healing *in vivo*. (A)** Representative immunofluorescence images of dorsal skin wound tissues from mice 3 days after wounding with/without EF treatment. Seven μm thick tissue sections were co-stained with F4/80 (showing in green in merged images), iNOS (magenta), and CD206 (red) as indicated. Nuclei were labeled with DAPI (blue). n = 13 mice for the control, and n = 9 for the EF treated group. 5 images were taken and quantified for each

wound. **(B)** Image of histomorphometric analysis of wound re-epithelization. Representative wounds: Green line indicates area of wound re-epithelialization in control and EF treated wounds. **(C)** M1 to M2 ratios from experiments shown in A and B. Plot shows mean ± standard error. **(D)** Quantitation of percent re-epithelialization. P values and percentage of changes from the control (CTRL) are as indicated. Plot shows mean ± standard error.

beneficial effects in the entire wound healing process. These are preliminary results and additional future studies using keratin 14 staining to evaluate keratinocytes as well as other markers of wound healing such as pro-inflammatory cytokines either as immunohistochemistry or RNA or Western analysis of the skin samples would will be beneficial to confirm the conclusions of this work [40, 41].

## Materials and methods

### Device fabrication process

The bioelectronic device consists of a Polydimethylsiloxane (PDMS) body with printed circuit board (PCB) integration. The PDMS body has two layers that have been bonded together. The top layer has four reservoirs and the bottom layer acts as a lid to enclose the reservoirs. The reservoirs are filled with Steinberg solution and integrated with AgCl wires needed to manipulate the targeted biological processes. The PCB is programmed to produce desired voltage and duration time of actuation. A 3V battery powers the PCB such that the EF treatment is actuated wirelessly without an external connected power source like in previous experiments.

### PCB board design

The control PCB is preprogrammed for timed delivery of the EF. A 3V battery powers the PCB such that the EF treatment is actuated wirelessly without an external connected power source and without limiting the movement of the animals. The control PCB consists of a low-power microcontroller that provides programmable electrical signals and takes advantage of deep-sleep mode to prolong battery run-time. The control PCB is electrically and mechanically integrated with the bonded PDMS piece using metallic pins that are coated with silver epoxy. To deliver an EF, a center reference electrode (negative relative to other electrodes) with four surrounding working electrodes (positive) is set up to establish an EF directed toward the center (the cathode) of the wound.

### Capillary tubing preparation

An I.D. 100 μm, O.D. 375 μm capillary tube was first cut into a 10 cm length and attached to a 3 mL plastic syringe with a Luer-to-Microtight adaptor (P-662, IDEX, USA). The capillary tubing was modified with several solutions for sequential etching, adhesion promotion, and cleaning processes using a syringe pump set at a flow rate of 5 to 7 μL/min. First, the inner surface was treated with 1.0 M sodium hydroxide (NaOH) solution for 24 hours to form hydroxyl groups. After 24 hours, the inner tubing surface was cleaned with deionized (DI) water and modified again with 3-(Trimethoxysilypropyl methacrylate (Silane A174) to form methacrylate groups on the surface by covalent bonding between the silanol and hydroxyl groups. After cleaning with ethanol, the modified capillary tubing was filled with a hydrogel solution made of 1.0 M 2-acrylamido-2-methyl-1-propanesulfonic acid (AMPSA), 0.4 M Polyethylene glycol diacrylate (PEDGA 575), and 0.05 M 1.1.2-hydroxy-4'-(2-hydroxyethoxyl)-2-methylpropiophenone (IRGACURE-2959 photoinitiator) in water. The hydrogel solution was cross-linked inside the tubing by exposure to UV-light for 5 minutes at 8 mJ/cm$^2$ with a mask aligner

(Hybralign 200IR, OAI, USA). The hydrogel-filled capillary tubing was preserved in DI water before assembling.

## Ionic conductivity measurement for quality control

Quality control ensures the best capillaries are chosen by measuring their ionic conductivity, which shows a consistent flow of ions between 10–20 µA. The test is done using a capillary treated with NaOH and filled with hydrogel for 5 minutes on the AutoLab. The setup includes a source reservoir of 0.1 M HCl and a target reservoir of 0.01 M KCl, with an applied voltage of 0.8 V. This allows ions to move from the source to the target reservoir.

## Animal wounding and daily treatment

The animal wounding surgery was performed under a protocol reviewed and approved by the UC Davis Institutional Animal Care and Use Committee (IACUC). C57BL/6J mice (males, >7-months old and >30 grams to carry the device) were purchased from the Jackson Lab (Sacramento, CA) and acclimated at least 1 week before the surgery. On Day 0, the mice were anesthetized (1–5% of isoflurane inhalation), shaved and recorded their weight to be randomly assigned to the treatment groups. Two 6 mm circular wounds were generated on the back by removing full thickness of skin with biopsy punch, surgical scissors and forceps. Silicon splint rings were sutured around the wounds to avoid skin contraction during healing [32, 42, 43]. Splinting the mouse wound allows the repair process to be dependent on epithelialization, cellular proliferation, and angiogenesis [44]. The sutured splints cause no secondary damage to the wound bed because the skin is not attached firmly to the muscle layer underneath [45, 46]. Instead, it stabilizes the wound and provides a platform to maintain the bioelectronic device localized to the wound site. The bioelectronic device or a mock PDMS control device rest on top of the splint rings allowing only the actuating bottom surface of the device to come in contact with the wound bed. Both the wounds on the mouse received the same treatment, either control or EF. Tegaderm dressing (3M, Maplewood, MN) maintains the device stationary to the wound preventing any slippage that can result in treatment failure. The post-op analgesic (buprenorphine 0.05–0.1 mg/kg, twice on the surgery day) was provided. The electric field stimulation (targeted to the strength of 125mV/mm) was set for 250 min per day while the animals were fully recovered from the anesthesia and free to move in the cage, and the device is required to re-start by battery change every day. During the post-op observation period, the animals were anesthetized briefly to measure weight, image the wounds, and re-start the device. The currents of each device were examined again after daily battery changes, and the Tegaderm dressings were re-applied. Animals were excluded from the analysis if the total anesthesia time was prolonged (over 100 minutes), or if an EF device fails to deliver current during the experiment period.

## Histologic analysis

On post-op day 3, mice were euthanized by cervical dislocation under deep anesthesia, and the wounds and the surrounding tissue were excised and fixed in a 4% paraformaldehyde solution for 24 hours. The fixed tissues were processed in a Tissue-Tek VIP 6 processor (Sakura Finetek, Torrance, CA, USA) and embedded in paraffin wax blocks. The wound tissue was sectioned into 5-um thickness, mounted on glass slides, stained with hematoxylin/eosin (H&E staining), and used to quantify the re-epithelialization rates.

For re-epithelialization, measurements were performed as previously described [32, 43]. Briefly, H&E stained sections were imaged on a BioRevo BZ-9000 inverted microscope, and scored using BZ Analyzer software (Keyence, Osaka, Japan). The left and right-wound edges

were identified by the innermost follicle present on each side, and epithelial ingrowth was measured for each side of the wound along the basal keratinocyte layer from the innermost follicle to the tip of the epithelial tongue. The total wound width was determined by measuring along the surface of the granulation tissue between the two follicles, and percent re-epithelialization was calculated as

$$\frac{left epithelial tongue length (um) + right epithelial tongue length (um)}{wound width (um)} \times 100.$$

## M1/M2 Immunohistochemical (IHC) staining

Formalin-fixed paraffin-embedded tissue samples were treated for antigen retrieval and blocked for 2 hours with 10% Donkey Serum (Thermo Fisher). They were then incubated overnight with primary antibodies specific for F4/80 (dilution 1:50; MCA497G, BIO-RAD, Hercules, CA), iNOS (PA3-030A, Thermo Fisher Scientific), and CD206 (PA5-46994, Thermo Fisher Scientific). The samples were then stained with Alexa Fluor-conjugated secondary antibodies (Donkey Anti rat-AlexaFluor 488, Donkey Anti rabbit-AlexaFluor 647, Donkey Anti goat-AlexaFluor 568, dilution 1:200, Thermo Fisher Scientific) and counterstained with DAPI. The samples were mounted with an anti-fade mountant (SlowFade Mountant; S36936, Thermo Fisher Scientific) and imaged using a high-resolution microscope (BZ-X800, KEYENCE, Itasca, IL). For each sample, we imaged 5 randomly selected fields at the wound center, which was identified by the absence of epidermal structures. The images were processed using ImageJ and CellProfiler 4.2 software for semi-quantification of the different macrophage subtypes. Wound beds were identified with a bright field view between the epithelial edges and confirmed with parallel H&E staining of neighbor sections (Fig 3). Five higher magnification images (40X objective) were taken with equal spacing between the epidermal tongue. The numbers of macrophage subtypes were manually counted based on double-positive staining by a researcher who was blinded to the treatment or control group of the sample being analyzed.

## Statistical analysis

Statistical analysis was carried out using a 2-tailed Student's t-test for heteroscedastic data. Results were considered significant at $P < 0.05$.

## Supporting information

**S1 File.** Contains S1 Table. Electric field device quality control check on 4.9.22, S2 Table. Electric field delivery experiment on 4.10.22–4.14.22, S3 Table. Test animal identification and biocompatibility evaluation, S1 Fig. Three iterations of microcontroller design, S2 Fig. Testing version 2 device current connection of electric field treatment using multimeters, S3 Fig. Bioelectronic device circuit system, S4 Fig. Re-epithelialization analysis, S5 Fig. Electrical stimulation modulated macrophage percentages during wound healing *in vivo*, S6 Fig. Experiment 20—High dose electric field wound treatment via device.
(DOCX)

## Acknowledgments

H.-C. H. would like to thank Vincent Pham and Harrison Shawa for assisting the animal surgery process.

## Author Contributions

**Conceptualization:** Roslyn Rivkah Isseroff, Marco Rolandi.

**Investigation:** Cristian O. Hernandez, Hao-Chieh Hsieh, Kan Zhu, Houpu Li, Hsin-ya Yang, Cynthia Recendez, Narges Asefifeyzabadi, Tiffany Nguyen, Maryam Tebyani, Prabhat Baniya, Moyasar A. Alhamo, Anthony Gallegos, Cathleen Hsieh, Alexie Barbee, Jonathan Orozco, Yao-Hui Sun.

**Methodology:** Hsin-ya Yang, Andrea Medina Lopez, Roslyn Rivkah Isseroff.

**Project administration:** Elham Aslankoohi.

**Supervision:** Hao-Chieh Hsieh, Athena M. Soulika, Mircea Teodorescu, Marcella Gomez, Narges Norouzi, Min Zhao, Marco Rolandi.

**Writing – original draft:** Cristian O. Hernandez, Hao-Chieh Hsieh, Kan Zhu, Houpu Li.

**Writing – review & editing:** Marcella Gomez, Min Zhao, Marco Rolandi.

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
