## [Decision Letter · Decision Letter 0]

6 Oct 2023

PONE-D-23-22607A bioelectronic device for treatment of wounds improves outcomes in an in vivo mouse modelPLOS ONE

Dear Dr. Rolandi,

Thank you for submitting your manuscript to PLOS ONE. After careful consideration, we feel that it has merit but does not fully meet PLOS ONE’s publication criteria as it currently stands. Therefore, we invite you to submit a revised version of the manuscript that addresses the points raised during the review process.

We look forward to receiving your revised manuscript.

Kind regards,

Abeer El Wakil, PhD

Academic Editor

PLOS ONE

Journal Requirements:

"This project is supported by the Defense Advanced Research Projects Agency (DARPA) through Cooperative Agreement Number D20AC00003 awarded by the U.S. Department of the Interior (DOI)."

"This project is supported by the Defense Advanced Research Projects Agency (DARPA) through Cooperative Agreement Number D20AC00003 awarded by the U.S. Department of the Interior (DOI), Interior Business Center. H.-C. H. would like to thank Vincent Pham and Harrison Shawa for assisting the animal surgery process."

"This project is supported by the Defense Advanced Research Projects Agency (DARPA) through Cooperative Agreement Number D20AC00003 awarded by the U.S. Department of the Interior (DOI)."

4. We note that Figures 1, 2, S1 and S2 in your submission contain copyrighted images. All PLOS content is published under the Creative Commons Attribution License (CC BY 4.0), which means that the manuscript, images, and Supporting Information files will be freely available online, and any third party is permitted to access, download, copy, distribute, and use these materials in any way, even commercially, with proper attribution. For more information, see our copyright guidelines: http://journals.plos.org/plosone/s/licenses-and-copyright.

a. You may seek permission from the original copyright holder of Figures 1, 2, S1 and S2 to publish the content specifically under the CC BY 4.0 license. 

Additional Editor Comments:

The concept of the present study seems interesting as the authors present a bioelectronic device for assisting in the treatment of wounds that possesses wearable and programmable properties. However, the manuscript in its current form is not acceptable for publication, as it needs improvement following the reviewers' concerns before its acceptance for publication.

Reviewer #1

In this work, the authors present a bioelectronic device for assisting in the treatment of wounds that possesses wearable and programmable properties. The bioelectronic device achieves modulation of wound regenerative cell migration and macrophage phenotype and differentiation by applying a programmable electric field to a mouse wound, leading to wound re-epithelialization and healing. This wearable electronic device proposed by the authors not only supports the application of programmed applied electric fields for in vivo wound healing, but also provides new ideas and a basis for the development of further techniques based on the modulation of macrophages and inflammation for better wound healing. The content of this work is interesting and deserves to be studied in depth. However, I have some concerns about the current version of this work.

1. Generally speaking, the wound surface has irregularity, and sewing often destroys part of the wound surface in vivo, which will easily cause the device to cause secondary damage to the wound. In addition, skin wounds tend to exude tissue fluid at the wound surface, and the effect of tissue fluid on the bioelectronic device is also considered and eliminated during biological experiments.

2. In this bioelectronic device, hydrogel is applied as the interface material between the device and the wound, and the solubilizing absorption and enrichment effect of hydrogel also promotes wound healing, does this have a superimposed effect with the wound healing effect of the bioelectronic device? If the stimulatory effect of an applied electric field on wound healing is explored, it is suggested that a control group where the electronic device is installed but no electric field is applied could be added for analysis.

3. The selected experimental items and control data are small, and the treatment test with three days as the cycle lacks certain rigor and persuasiveness, the results of a longer cycle can be taken for observation and analysis. At the same time, the healing effect of mouse wounds should be tested by providing some data on the change of wound size, not only the wound re-epithelialization rate and macrophage phenotype percentage of the wound, so as to intuitively correspond to the incentive effect of wounds.

4. What is the basis for choosing an applied electric field of 125 ± 75 mv/mm for the electronic device? Some prospective analysis can be made of the effect of the parameters associated with this bioelectronic device on wound healing in vivo and the optimization of the associated parameters.

5. Some of the pictures are not clearly drawn and contain logical errors. The drawing of Figure 2C is inconsistent with the contextual description, and the picture of electronic equipment in Figure S2 in the supporting information does not match the previous discussion. Please check carefully.

6. The format of some references is not standardized.

Reviewers' comments:

Reviewer's Responses to Questions

**Comments to the Author**

1. Is the manuscript technically sound, and do the data support the conclusions?

Reviewer #1: Yes

2. Has the statistical analysis been performed appropriately and rigorously? 

Reviewer #1: Yes

3. Have the authors made all data underlying the findings in their manuscript fully available?

Reviewer #1: Yes

4. Is the manuscript presented in an intelligible fashion and written in standard English?

Reviewer #1: Yes

5. Review Comments to the Author

Reviewer #1: In this work, the authors present a bioelectronic device for assisting in the treatment of wounds that possesses wearable and programmable properties. The bioelectronic device achieves modulation of wound regenerative cell migration and macrophage phenotype and differentiation by applying a programmable electric field to a mouse wound, leading to wound re-epithelialization and healing. This wearable electronic device proposed by the authors not only supports the application of programmed applied electric fields for in vivo wound healing, but also provides new ideas and a basis for the development of further techniques based on the modulation of macrophages and inflammation for better wound healing. The content of this work is interesting and deserves to be studied in depth. However, I have some concerns about the current version of this work.

1. Generally speaking, the wound surface has irregularity, and sewing often destroys part of the wound surface in vivo, which will easily cause the device to cause secondary damage to the wound. In addition, skin wounds tend to exude tissue fluid at the wound surface, and the effect of tissue fluid on the bioelectronic device is also considered and eliminated during biological experiments.

2. In this bioelectronic device, hydrogel is applied as the interface material between the device and the wound, and the solubilizing absorption and enrichment effect of hydrogel also promotes wound healing, does this have a superimposed effect with the wound healing effect of the bioelectronic device? If the stimulatory effect of an applied electric field on wound healing is explored, it is suggested that a control group where the electronic device is installed but no electric field is applied could be added for analysis.

3. The selected experimental items and control data are small, and the treatment test with three days as the cycle lacks certain rigor and persuasiveness, the results of a longer cycle can be taken for observation and analysis. At the same time, the healing effect of mouse wounds should be tested by providing some data on the change of wound size, not only the wound re-epithelialization rate and macrophage phenotype percentage of the wound, so as to intuitively correspond to the incentive effect of wounds.

4. What is the basis for choosing an applied electric field of 125 ± 75 mv/mm for the electronic device? Some prospective analysis can be made of the effect of the parameters associated with this bioelectronic device on wound healing in vivo and the optimization of the associated parameters.

5. Some of the pictures are not clearly drawn and contain logical errors. The drawing of Figure 2C is inconsistent with the contextual description, and the picture of electronic equipment in Figure S2 in the supporting information does not match the previous discussion. Please check carefully.

6. The format of some references is not standardized.

6. PLOS authors have the option to publish the peer review history of their article (what does this mean?). If published, this will include your full peer review and any attached files.

Reviewer #1: No

---

## [Decision Letter · Decision Letter 1]

18 Jan 2024

PONE-D-23-22607R1A bioelectronic device for electric field treatment of wounds improves outcomes in an in vivo mouse modelPLOS ONE

Dear Dr. Rolandi,

Thank you for submitting your manuscript to PLOS ONE. After careful consideration, we feel that it has merit but does not fully meet PLOS ONE’s publication criteria as it currently stands. Therefore, we invite you to submit a revised version of the manuscript that addresses the points raised during the review process.

Please submit your revised manuscript by Mar 03 2024 11:59PM. If you will need more time than this to complete your revisions, please reply to this message or contact the journal office at plosone@plos.org. Please include the following items when submitting your revised manuscript:A rebuttal letter that responds to each point raised by the academic editor and reviewer(s). You should upload this letter as a separate file labeled 'Response to Reviewers'.A marked-up copy of your manuscript that highlights changes made to the original version. You should upload this as a separate file labeled 'Revised Manuscript with Track Changes'.An unmarked version of your revised paper without tracked changes. You should upload this as a separate file labeled 'Manuscript'.

We look forward to receiving your revised manuscript.

Kind regards,

Abeer El Wakil, PhD

Academic Editor

PLOS ONE

**Additional Editor Comments:**

The concerns raised by the reviewers were partly addressed. To be able to deliver a positive decision on the manuscript, the authors need to address all raised concerns.

Reviewers' comments:

Reviewer's Responses to Questions

**Comments to the Author**

1. If the authors have adequately addressed your comments raised in a previous round of review and you feel that this manuscript is now acceptable for publication, you may indicate that here to bypass the “Comments to the Author” section, enter your conflict of interest statement in the “Confidential to Editor” section, and submit your "Accept" recommendation.

Reviewer #2: All comments have been addressed

Reviewer #3: (No Response)

2. Is the manuscript technically sound, and do the data support the conclusions?

Reviewer #2: Yes

Reviewer #3: Partly

3. Has the statistical analysis been performed appropriately and rigorously? 

Reviewer #2: Yes

Reviewer #3: No

4. Have the authors made all data underlying the findings in their manuscript fully available?

Reviewer #2: Yes

Reviewer #3: Yes

5. Is the manuscript presented in an intelligible fashion and written in standard English?

Reviewer #2: Yes

Reviewer #3: Yes

6. Review Comments to the Author

Reviewer #2: The authors have thoroughly revised the manuscript ，and the manuscript is now on the sharp for publication.

Reviewer #3: Hernandez et al. extended their previous study (Ref#28) for a proof-of-principle application of the electric field (EF) treatment on wound healing in an in-vivo setting. For this, they used EF for 3 days to a surgical wound. Although conceptually novel and interesting, the data lack technical and methodological robustness (see below). Moreover, the authors did not perform any new experiments to address the concerns of the previous reviewer (comment#3), which were justified, relatively easy and feasible experiments. I have following comments.

1. In line #112-114, authors claimed a decreased in M1 macrophage counts and unchanged M2 macrophage counts in treated mice. To this reviewer the representative images showed a decreased total DAPI+ cells in treated mice including F4/80+ macrophages, M1 like macrophages, but without any change in M2 like cells compared to untreated mice. So, I agree with reviewer 1 the need to quantify macrophage phenotype percentages in addition to the M1/M2 ratio.

2. In line # 118-119, authors claimed increased M2 macrophages due to proliferation which is contrary to the statement in line # 112-114. There is a need to quantify proliferation and apoptosis for both macrophage phenotypes before making such claim.

3. In line # 127-129, authors claimed significantly increased re-epthelialization with a p-value above >0.05 which is against the any significant value in statistical analysis. This is a very premature claim. Moreover, authors should provide higher magnification images with clearly visible new-epithelial layers.

4. Statistics. It is not clear which student’s t-test was used and the rationale for this. In addition, I did not find the reason to mention about ANOVA in the method part. Moreover, there is inconsistency in the biological replicates presented in figure 3 and in the figure legend.

7. PLOS authors have the option to publish the peer review history of their article (what does this mean?). If published, this will include your full peer review and any attached files.

Reviewer #2: No

Reviewer #3: **Yes: **Sarajo Mohanta

---

## [Author Response · Author response to Decision Letter 1]

20 Feb 2024

The response to reviewers is attached as a Word file.

---

## [Decision Letter · Decision Letter 2]

1 Apr 2024

PONE-D-23-22607R2A bioelectronic device for electric field treatment of wounds improves outcomes in an in vivo mouse modelPLOS ONE

Dear Dr. Rolandi,

Thank you for submitting your manuscript to PLOS ONE. After careful consideration, we feel that it has merit but does not fully meet PLOS ONE’s publication criteria as it currently stands. Therefore, we invite you to submit a revised version of the manuscript that addresses the points raised during the review process.

We look forward to receiving your revised manuscript.

Kind regards,

Abeer El Wakil, PhD

Academic Editor

PLOS ONE

Journal Requirements:

Reviewers' comments:

Reviewer's Responses to Questions

**Comments to the Author**

1. If the authors have adequately addressed your comments raised in a previous round of review and you feel that this manuscript is now acceptable for publication, you may indicate that here to bypass the “Comments to the Author” section, enter your conflict of interest statement in the “Confidential to Editor” section, and submit your "Accept" recommendation.

Reviewer #4: (No Response)

Reviewer #5: (No Response)

2. Is the manuscript technically sound, and do the data support the conclusions?

Reviewer #4: Partly

Reviewer #5: Partly

3. Has the statistical analysis been performed appropriately and rigorously? 

Reviewer #4: No

Reviewer #5: Yes

4. Have the authors made all data underlying the findings in their manuscript fully available?

Reviewer #4: (No Response)

Reviewer #5: (No Response)

5. Is the manuscript presented in an intelligible fashion and written in standard English?

Reviewer #4: Yes

Reviewer #5: (No Response)

6. Review Comments to the Author

Reviewer #4: In this paper, the authors successfully prepared a standalone, wearable, and programmable electronic device to administer a well-controlled exogenous EF, aiming to accelerate wound healing in a mouse model in vivo. However, there are some major concerns and questions the authors should consider before publication:

Comment 1) In Figure 3D, the statistical result shows that the P-value above 0.05, then how to support the conclusion, this point seems to need to be explained.

Comment 2) In the in vivo part, why the sample number of different groups is inconsistent.

Comment 3) A good excellent academic paper should not only show the experimental results, but also give an in-depth discussion on these results. The author should pay attention to it. Some related studies (such as ACS Applied Materials & Interfaces 2022 14 (26), 29491-29505; Cell Mol Biol Lett. 2023 Jul 28;28(1):61, and 2024 Feb 5;29(1):24. can be cited and discussed to improve the discussion of this work.

Reviewer #5: This manuscript has novel findings and it may benefit in treating wounds. However, there is a major concern that this manuscript is based on just one significant result, which is the M1/M2 macrophage ratio. The percent re-epithelialization results were not significant in control versus EF treatment. The study lacks other criteria that may support the conclusions. Below are the concerns in detail:

Concerns:

1. The re-epithelialization were performed on H&E stained skin section which is not clear, however, the migrating keratinocytes can be better evaluated by keratin 14 staining which is the marker for keratinocytes or by keratin 17 staining as described in this publication (https://www.ncbi.nlm.nih.gov/pmc/articles/PMC4086220/ and https://www.sciencedirect.com/science/article/pii/S0022202X15327469).

2. Authors should show the other markers of wound healing such as pro-inflammatory cytokines (IL1, TNF-alpha, etc.) either as immunohistochemistry or RNA or Western analysis of the skin samples.

3. Figure 3, Panel B: It is not clear as what is on the right panel (I assume it is the zoom image of the left edge and right edge of the wound). This description can be in the legends. Also, on the right panel, it will be better if there is heading on the top (like, Left edge or Right Edge.

4. Figure 3 C and D: Please clearly state if the data presented is shown in SEM or SD for each of the panel. It is also better if authors can stick to SEM or SD in all the figures in this manuscript. Also, correct this information in the Methods for Statistical Analysis section.

5. Figure 3, panel A: Is there a reason behind keeping the iNOS alone, CD206 alone, F4/80 alone, and DAPI alone staining in the upper two rows in black and white? Why not a colored one?

6. Abstract, Line 26: It will be better to write “an in vivo mouse model” instead of “a mouse model in vivo”.

7. Results, Line 199: Please make it clear in the methods that both the wounds on single mouse received the same treatment (control or EF).

7. PLOS authors have the option to publish the peer review history of their article (what does this mean?). If published, this will include your full peer review and any attached files.

Reviewer #4: No

Reviewer #5: No

---

## [Author Response · Author response to Decision Letter 2]

26 Apr 2024

Response letter has been attached.

---

## [Editor Report · Decision Letter 3]

30 Apr 2024

A bioelectronic device for electric field treatment of wounds improves outcomes in an in vivo mouse model

PONE-D-23-22607R3

Dear Dr. Rolandi,

We’re pleased to inform you that your manuscript has been judged scientifically suitable for publication and will be formally accepted for publication once it meets all outstanding technical requirements.

Kind regards,

Abeer El Wakil, PhD

Academic Editor

PLOS ONE
---

## [Editor Report · Acceptance letter]

24 May 2024

PONE-D-23-22607R3 

PLOS ONE

Dear Dr. Rolandi, 

I'm pleased to inform you that your manuscript has been deemed suitable for publication in PLOS ONE. Congratulations! Your manuscript is now being handed over to our production team.

Kind regards, 

on behalf of

Professor Abeer El Wakil 

Academic Editor

PLOS ONE